# Bone Healing Materials in the Treatment of Recalcitrant Nonunions and Bone Defects

**DOI:** 10.3390/ijms23063352

**Published:** 2022-03-20

**Authors:** Emérito Carlos Rodríguez-Merchán

**Affiliations:** 1Department of Orthopedic Surgery, La Paz University Hospital—IdiPaz, 28046 Madrid, Spain; ecrmerchan@hotmail.com; 2Osteoarticular Surgery Research, Hospital La Paz Institute for Health Research—IdiPaz, 28046 Madrid, Spain

**Keywords:** bone defects, recalcitrant nonunions, bone healing materials, bone graft substitutes, bioactive molecules, stem cells

## Abstract

The usual treatment for bone defects and recalcitrant nonunions is an autogenous bone graft. However, due to the limitations in obtaining autogenous bone grafts and the morbidity associated with their procurement, various bone healing materials have been developed in recent years. The three main treatment strategies for bone defects and recalcitrant nonunions are synthetic bone graft substitutes (BGS), BGS combined with bioactive molecules, and BGS and stem cells (cell-based constructs). Regarding BGS, numerous biomaterials have been developed to prepare bone tissue engineering scaffolds, including biometals (titanium, iron, magnesium, zinc), bioceramics (hydroxyapatite (HA)), tricalcium phosphate (TCP), biopolymers (collagen, polylactic acid (PLA), polycaprolactone (PCL)), and biocomposites (HA/MONs@miR-34a composite coating, Bioglass (BG)-based ABVF-BG (antibiotic-releasing bone void filling) putty). Bone tissue engineering scaffolds are temporary implants that promote tissue ingrowth and new bone regeneration. They have been developed to improve bone healing through appropriate designs in terms of geometric, mechanical, and biological performance. Concerning BGS combined with bioactive molecules, one of the most potent osteoinductive growth factors is bone morphogenetic proteins (BMPs). In recent years, several natural (collagen, fibrin, chitosan, hyaluronic acid, gelatin, and alginate) and synthetic polymers (polylactic acid, polyglycolic acid, polylactic-coglycolide, poly(e-caprolactone) (PCL), poly-p-dioxanone, and copolymers consisting of glycolide/trimethylene carbonate) have been investigated as potential support materials for bone tissue engineering. Regarding BGS and stem cells (cell-based constructs), the main strategies are bone marrow stromal cells, adipose-derived mesenchymal cells, periosteum-derived stem cells, and 3D bioprinting of hydrogels and cells or bioactive molecules. Currently, significant research is being performed on the biological treatment of recalcitrant nonunions and bone defects, although its use is still far from being generalized. Further research is needed to investigate the efficacy of biological treatments to solve recalcitrant nonunions and bone defects.

## 1. Introduction

An autogenous bone graft is the safest and most effective grafting procedure for treating bone defects because it contains mesenchymal stem cells (MSCs) from the patient that enhance osteogenesis, as well as growth factors that enhance osteoinduction. Bone autografts also provide a calcified osteoconductive framework for new bone to grow [1,2,3,4,5,6,7].

However, due to the limitations in obtaining autogenous bone grafts and the morbidity associated with their procurement, various bone substitutes have been developed in recent years. Complications of autogenous bone graft procurement are usually minor, with a frequency of about 20%. However, these complications sometimes significantly affect the donor site and thus the patient (approximately 5%) [5,8,9].

Regarding the prevalence of the use of bone healing materials, European surveys reveal an increasing trend in the use of advanced therapy medicinal products (ATMPs) for bone regeneration (4% of the total use of ATMPs) [10].

In this article, treatment strategies for recalcitrant nonunions and bone defects using synthetic bone graft substitutes (BGS), BGS combined with bioactive molecules, and graft substitutes and stem cells (cell-based constructs) are reviewed (Figure 1).

## 2. Synthetic Bone Graft Substitutes (BGS)

Table 1 shows the strategies to heal recalcitrant nonunions and bone defects using synthetic BGS.

### 2.1. Biomaterials

Biomaterials are very important in scaffold-based bone tissue engineering. Therefore, numerous biomaterials have been developed to prepare bone tissue engineering scaffolds, including biometals (titanium (Ti) [16], iron (Fe) [17,18], magnesium (Mg) [19,20], zinc (Zn) [21]), bioceramics (hydroxyapatite (HA)) [22,23], tricalcium phosphate (TCP) [24], biopolymers (collagen [25], polylactic acid (PLA) [26], polycaprolactone (PCL) [27]), and biocomposites (HA/MONs@miR-34a composite coating, Bioglass (BG)-based ABVF-BG (antibiotic-releasing bone void filling) putty) [28,29].

### 2.2. Bone Tissue Engineering Scaffolds

Bone tissue engineering scaffolds are temporary implants that promote tissue ingrowth and new bone regeneration [30,31]. They have been developed to improve bone healing through appropriate designs in terms of geometric, mechanical, and biological performance [32,33].

Polycaprolactone (PCL) is a biocompatible polymeric material approved by the U.S. Food and Drug Administration [34]. Compared to other types of biomaterials, PCL exhibits high design flexibility at a low melting temperature, and slow biodegradation for long-term service [11]. However, PCL scaffolds have limitations on biofunctional sites, resulting in ineffective cellular responses [12].

In a study by Dong et al., cell-free Mg-incorporated PCL-based scaffolds were prepared by 3D printing for bone healing [13]. The Mg microparticles endowed these scaffolds with good physical, chemical, and bioactive properties. In vitro and in vivo experiments showed that the Mg/PCL scaffolds exhibited good biocompatibility, enhanced osteogenic and angiogenic activity, and a good ability to form new bone [13].

Wiese and Pappe developed 3D printed customized Mg/PCL composite scaffolds with enhanced osteogenesis and biomineralization for the treatment of bone defects caused by high-energy injuries, or by bone loss or infections [14]. Mg microparticles embedded in such scaffolds played a positive role in enhancing biocompatibility, biomineralization, and biodegradable capabilities. When incorporated with 3% Mg, the PCL scaffolds showed optimal bone repair capabilities in vitro and in vivo. In vitro experiments indicated that the 3% Mg/PCL scaffolds had adequate mechanical properties, good biocompatibility, and good osteogenic and angiogenic activities. In addition, in vivo studies demonstrated that Mg/PCL scaffolds promoted tissue ingrowth and new bone formation. This study found that 3D printed cell-free Mg/PCL scaffolds are promising for bone healing [14].

Recent advances in the fabrication of nanoscale metal–organic framework (nano-MOF) scaffolds have made it possible to enhance the properties of scaffolds in bone tissue engineering [15].

## 3. BGS Combined with Bioactive Molecules

Table 2 summarizes the strategies for the treatment of bone defects and recalcitrant nonunions using BGS combined with biologically active substances.

### 3.1. Bone Morphogenetic Proteins (BMPs)

One of the most potent osteoinductive growth factors is BMPs [44]. These multifunctional cytokines are involved in all stages of fracture healing by inducing the differentiation of MSCs into chondrogenic and osteogenic lineages, stimulating angiogenesis, and increasing alkaline phosphatase activity [35]. BMPs are non-collagenous low-molecular-weight glycoproteins that belong to the transforming growth factor-beta TGF superfamily [45]. Although more than 20 homodimeric or heterodimeric BMPs have been identified, only a few members of this family (BMP-2, -3, -6, -7, and -9) are truly osteogenic [35]. The osteogenic potential of BMPs was first discovered by Urist in 1965, when he demonstrated the induction of bone formation after implanting a demineralized bone matrix in ectopic sites in rats [46].

Only BMP-2 (Infuse^®^ bone graft) and BMP-7 (OP-1 putty^®^), also known as osteogenic protein 1 (OP-1), have been approved by the FDA for the treatment of very specific bone fractures that exhibit delayed or incomplete healing. However, OP-1 putty was withdrawn from the market in 2014 [44]. Infuse is expensive and requires supraphysiologic concentrations (10–1000 times higher) to induce bone healing. Infuse uses absorbable collagen sponges (ACS) and putty collagen particles [44]. The use of a natural component such as collagen for this purpose appeared to be very promising due to its biodegradability, biocompatibility, and ability to support mineralization and cell ingrowth in an osteoconductive manner [40]. However, there are several serious side effects in clinical applications associated with this carrier system. Placement of the ACS during surgery is often difficult, and, in some cases, secondary displacement of the collagen sponge occurs [41].

The main problem associated with ACS is the initial release of BMP-2 into the local environment due to the low binding affinity of BMPs for collagen [47]. This can produce heterotopic ossification in muscle tissue, as surrounding mesenchymal lineage progenitor cells in adjacent muscle differentiate into osteoblasts and cause mineral deposition under BMP stimulation [36]. Elevated levels of BMP-2 can also activate osteoclasts, leading to bone resorption [37]. In spine surgery, side effects such as inflammation, bone cysts, and neurological impairment have been reported [38]. It is therefore necessary to provide a suitable delivery system or vehicle for the controlled and continuous release of BMPs.

### 3.2. Alternative Carriers for Growth Factor Delivery

The major challenge when trying to develop osteoinductive bone grafts is the short systemic half-life of growth factors in the bloodstream, as with BMPs, which are rapidly degraded by proteinases within 7 to 16 min [38]. Supraphysiological concentrations of BMPs are used in an attempt to compensate for the short retention time in the defect zone and to enhance signaling efficiency, but they can produce severe side effects due to a burst release [38]. Therefore, biomaterials are needed that allow a sustained spatiotemporal release of growth factors, ensure a prolonged presence of BMPs at the implantation site, prevent their systemic diffusion, and maintain their local concentrations at a constant physiological level [35].

The requirements to be met by the ideal growth factor carrier are enormous, and, to date, no carrier really meets all expectations. A highly porous and osteoconductive three-dimensional carrier structure must have adhesion sites for cell ligands, contain affinity motifs for growth factor binding, fill the defect, and have adequate mechanical properties. It must be biocompatible and biodegradable while protecting the BMPs from degradation. In addition, it should be non-toxic, non-allergenic, non-carcinogenic, easily sterilizable, stable, and cost effective [48].

In recent years, several natural and synthetic polymers have been investigated as potential support materials for bone tissue engineering [49]. Natural polymers such as collagen, fibrin, chitosan, hyaluronic acid, gelatin, and alginate have clear advantages due to their inherent biocompatibility and bioactivity, but they lack the mechanical properties necessary for load-bearing applications. In addition, they have fixed degradation rates, are difficult to harvest and sterilize, and exhibit batch-to-batch variability. In some cases, they can transmit pathogens and induce an immunogenic response [39].

Synthetic polymers have a defined chemistry and are tunable in terms of porosity and degradation time, but they lack inherent bioactivity. They can be produced in large quantities under controlled conditions, have a long shelf life, are easy to process, and are often cheaper than biological scaffolds [50]. The most commonly used synthetic polymers in tissue engineering applications with FDA approval and in clinical applications are aliphatic polymers such as polylactic acid (PLA), polyglycolic acid (PGA), polylactic-coglycolide (PLGA), poly(e-caprolactone) (PCL), poly-p-dioxanone, and copolymers consisting of glycolide/trimethylene carbonate [50].

### 3.3. Small Molecules as Regulators of Bone Mass

Another option is to use small molecule bone mass regulators [51].

#### 3.3.1. Parathyroid Hormone (PTH)

PTH plays a key role in the regulation of calcium phosphate metabolism. Its production increases in response to low serum calcium levels. In addition, PTH potentiates the Wnt-beta catenin pathway, which is essential for osteogenesis and bone formation. It is also used as a drug to treat osteoporosis. The products developed by Kuros (KUR-111/112/113) contain PTH trapped in a natural fibrin matrix combined with a structural ceramic component (HAP/TCP granules), which provides mechanical stability during bone healing. The bioactive products are based on an engineered active fragment of human PTH linked to a transglutaminase substrate for binding to fibrin as a delivery mechanism, and a cell invasion matrix with an intervening plasmin-sensitive link. PTH was initially tested in femur and humerus defects of female sheep, where it was shown to be both osteoconductive and osteoinductive [42].

#### 3.3.2. KUR-111, KUR-112, and KUR-113

KUR-111 is a bone graft substitute initially developed for the treatment of tibial plateau fractures, where it was successful in a Phase IIb clinical study (NCT00533793, completed in 2011). This study evaluated the safety and efficacy of KUR-111 in 183 patients from 30 centers in Europe and Australia. At 16 weeks, 84% of the patients treated with an autograft and 84% of the patients treated with the highest dose of KUR-111 had radiological fracture healing. KUR-113 was developed for fractures at risk of incomplete healing. It was initially tested in tibial shaft fractures in a Phase II clinical trial and later for spinal fusion in patients with degenerative disc disease. KUR-112 is a drug candidate for patients with solitary bone cysts [43].

In summary, several bioactive molecules have been used to enhance bone repair. Full-length growth factors or protein-derived peptides appear to be the most promising strategies. It should be noted that the carriers used thus far to deliver these bioactive molecules are mainly extracellular matrix proteins (collagen, fibrin), or biomimetic calcium phosphate, which provide osteoconduction and mechanical support.

## 4. Bone Graft Substitutes and Stem Cells (Cell-Based Constructs)

The strategies to heal bone defects and recalcitrant nonunions using BGS and stem cells (cell-based constructs) are summarized in Table 3.

### 4.1. Bone Marrow Stromal Cells (BMSCs)

BMSCs represent a heterogeneous cell population that can be harvested from the stromal fraction of the bone marrow [58]. This harvesting is generally carried out by means of bone marrow aspiration from the wing of the ilium, the medial part of the proximal tibia, and/or the proximal humerus.

Although the extracellular matrix present in marrow is scarce, gentle mechanical disruption can easily dissociate stroma and hematopoietic cells from the bone marrow harvested into a single-cell suspension [59]. From an aspirate of 400–500 mL of bone marrow, approximately 100,000–130,000 BMSCs can be obtained [60]. Injection of BMSCs into a stabilized fracture appears to contribute to direct ossification [52].

### 4.2. Adipose-Derived Mesenchymal Cells (ASCs)

ASCs are a heterogeneous cell population that has been intensively studied for regenerative purposes [61]. This progenitor cell population has shown multilineage differentiation potential including adipogenic, osteogenic, chondrogenic, and myogenic lineages [62]. The attractive properties of ASCs are due to the fact that adipose tissue is abundant, easy to harvest, and present in various types of adipose tissue, including visceral fat, subcutaneous fat, and organ fat. Adipose obtained by liposuction is especially convenient since the procedure provides homogenous finely minced fragments that can be easily enzymatically digested in a short time. Approximately 1 g of adipose tissue can yield about 2 million cells, of which 10% are ASCs [63,64].

After isolation, ASCs adhere to plastic and can expand in vitro with a 10-fold higher CFU-F (colony-forming unit-fibroblastic) unit than BMSCs, with a rapid population doubling time of approximately 60 h [64]. This interesting cell population appears to be immune-privileged, given it appears to be protective against acute graft-versus-host disease [64]. ASCs also exert an immunosuppressive effect by inhibiting the proliferation of activated allogenic lymphocytes, which could be an attractive feature if used for allogenic implants [65,66].

In preclinical studies, ASCs were seeded onto scaffolds for critical-size mouse calvarial defects, demonstrating significant intramembranous bone formation and areas of complete bone regeneration, with a contribution of 84–99% from the implanted cells [53]. ASCs have also shown promising results for spinal fusion, with lower infiltration of inflammatory cells along with superior fusion compared to scaffolds without ASCs [54,55]. Although promising data have been published, sporadic publications on the tumor-enhancing properties of ASCs suggest caution in their clinical applications. Furthermore, significant differences have been reported with respect to the differentiation capacity of isolated ASCs according to their different anatomical locations, and according to the age and sex of the donors [67,68].

Finally, the key transcription factors and molecular events that steer ASC differentiation are largely unknown. Thus, ASCs might have potential as an attractive cell source for bone regeneration purposes and could be suitable for stabilized fractures with intramembranous bone healing. However, a comprehensive and systematic evaluation of in vitro expanded ASCs—with respect to their safety, reproducibility, and clinical quality—is needed.

### 4.3. Periosteum-Derived Stem Cells (PDSCs)

The periosteum is a thin 100 mm membrane that envelops all external bone surfaces not covered by cartilage [69]. This membrane is composed of an outer fibrous layer (70 mm) adjacent to the surrounding fibrous and muscular tissue, and an inner cambium layer (30 mm) [70]. The inner cambium layer is directly connected to the outer bone cortex, is highly vascularized, and serves as a host for osteochondroprogenitor cells with a unique tissue-building capacity [71]. In healthy bone development and homeostasis, cambium layer cells give rise to osteoblasts, allowing appositional bone growth and remodeling in concert with osteoclasts [72]. In a bone fracture, periosteal progenitor cells undergo massive expansion and subsequent chondrogenic and/or osteogenic differentiation [73]. This process leads to callus development, which further drives fracture healing.

Periosteal tissue can be obtained surgically from a patient by using a periosteal elevator, which cuts the fibers that anchor the periosteum to the bone [74]. From a biopsy of 1 g of periosteum, approximately 150,000 skeletal progenitor cells can be isolated for in vitro expansion. PDSCs can be enzymatically released from the periosteal biopsy onto plastic culture dishes where they have shown single-cell-derived clonal populations [75]. Under serum-containing conditions, human PDSCs (hPDSCs) have shown an in vitro expansion potential of up to 30 population doublings, with cells showing a fibroblast-like morphology and a population doubling time of approximately 55 h [56].

### 4.4. Three-Dimensional (3D) Bioprinting of Hydrogels and Cells or Bioactive Molecules

Three-dimensional printed scaffolds have been shown to promote tissue repair. However, the cell-level specific regulatory network activated by 3D printed scaffolds with various material components to form a symbiosis niche is not known. In 2022, Ji et al. fabricated three typical 3D printed scaffolds to explore the regulatory effect of the symbiotic microenvironment during bone healing: a natural polymer hydrogel (gelatin-methacryloyl, GelMA), a synthetic polymer material (polycaprolactone, PCL), and a bioceramic (tricalcium phosphate (TCP)) [57]. Enrichment analyses showed that the hydrogel promoted tissue regeneration and reconstruction by enhancing blood vessel generation through improved oxygen transport and the development of red blood cells. The PCL scaffold regulated cell proliferation and differentiation, promoted cell senescence, cell cycle, and deoxyribonucleic acid (DNA) replication pathways, and accelerated the process of endochondral ossification and callus formation. The TCP scaffold was able to specifically enhance the expression of genes of the osteoclast differentiation pathway and the extracellular space to promote osteoclast differentiation and favor the process of bone remodeling [57].

## 5. Conclusions

Regarding synthetic bone graft substitutes (BGS), biomaterials are paramount in scaffold-based bone tissue engineering. Bone tissue engineering scaffolds are temporary implants that promote tissue ingrowth and new bone regeneration. In vitro and in vivo experiments have shown that Mg/PCL (magnesium/polycaprolactone) scaffolds exhibit good biocompatibility, enhanced osteogenic and angiogenic activity, and a good ability to form new bone. Three-dimensional printed customized Mg/PCL composite scaffolds with enhanced osteogenesis and biomineralization have been used for the treatment of bone defects caused by high-energy injuries, or by bone loss or infections. Nanoscale metal–organic framework (nano-MOF) scaffolds have made it possible to enhance the properties of scaffolds in bone tissue engineering.

Concerning BGS combined with bioactive molecules, side effects such as inflammation, bone cysts, and neurological impairment have been reported in spine surgery following treatment with BMPs. Several natural and synthetic polymers have been investigated as potential support materials for bone tissue engineering. The main natural polymers are collagen, fibrin, chitosan, hyaluronic acid, gelatin, and alginate. The main synthetic polymers in tissue engineering applications with FDA approval and in clinical applications are aliphatic polymers such as PLA, PGA, PLGA, poly(e-caprolactone) (PCL), poly-p-dioxanone, and copolymers consisting of glycolide/trimethylene carbonate. A preclinical study on lumbar interbody spinal fusion concluded that peptide-coated granules produced more bone than in the control group without producing heterotopic ossification. P-15, a 15-amino acid peptide derived from collagen, has been used to treat fractures with delayed union in patients. Histological evaluation of the fracture callus resulted in encouraging clinical and radiographic outcomes. PTH was initially tested in femur and humerus defects of female sheep, where it was shown to be both osteoconductive and osteoinductive. KUR-113 was initially tested in tibial shaft fractures in a Phase II clinical trial and later for spinal fusion in patients with degenerative disc disease. In preclinical studies, ASCs were seeded onto scaffolds for critical-size mouse calvarial defects, demonstrating significant intramembranous bone formation and areas of complete bone regeneration, with a contribution of 84–99% from the implanted cells. ASCs have also shown promising results for spinal fusion.

Regarding bone graft substitutes and stem cells (cell-based constructs), a tricalcium phosphate (TCP) scaffold was able to specifically enhance the expression of genes of the osteoclast differentiation pathway and the extracellular space to promote osteoclast differentiation and favor the process of bone remodeling.

## Figures and Tables

**Figure 1 ijms-23-03352-f001:**
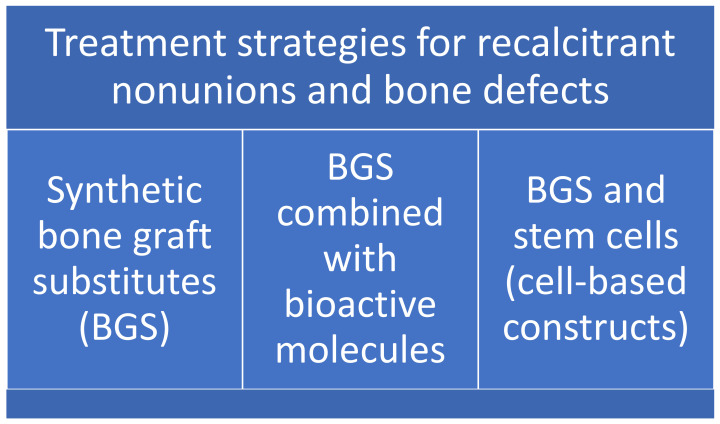
Main current strategies to treat bone defects and recalcitrant nonunions.

**Table 1 ijms-23-03352-t001:** Strategies to heal bone defects and recalcitrant nonunions using synthetic bone graft substitutes (BGS).

Strategy	Advantages and Disadvantages
**BIOMATERIALS** Titanium; iron; magnesium; zinc; bioceramics (hydroxyapatite); tricalcium phosphate; biopolymers (collagen, polylactic acid, polycaprolactone (PCL)); and biocomposites (HA/MONs@miR-34a composite coating, Bioglass (BG)-based ABVF-BG (antibiotic-releasing bone void filling) putty).	Compared to other types of biomaterials, polycaprolactone (PCL) exhibits high design flexibility at a low melting temperature and slow biodegradation for long-term service [11]. However, PCL scaffolds have limitations on biofunctional sites, resulting in ineffective cellular responses [12].
**THREE-DIMENSIONAL (3D) PRINTED SCAFFOLDS** Three-dimensional printing to customize the design of patient-specific BGS, and 3D printed cell-free Mg (magnesium)/PCL (polycaprolactone) scaffolds.	Mg/PCL scaffolds exhibited good biocompatibility, enhanced osteogenic and angiogenic activity, and a good ability to form new bone [13]. In vitro experiments indicated that the 3% Mg/PCL scaffolds had adequate mechanical properties, good biocompatibility, and good osteogenic and angiogenic activities. In addition, in vivo studies demonstrated that Mg/PCL scaffolds promoted tissue ingrowth and new bone formation. A study found that 3D printed cell-free Mg/PCL scaffolds are promising for bone healing [14].
**NANOSCALE METAL–ORGANIC FRAMEWORK (NANO-MOF)** **SCAFFOLD**	Recent advances in the fabrication of nanoscale metal–organic framework (nano-MOF) scaffolds have made it possible to enhance the properties of scaffolds in bone tissue engineering [15].

**Table 2 ijms-23-03352-t002:** Strategies to heal bone defects using BGS combined with biologically active substances.

Strategy	Advantages and Disadvantages
**BONE MORPHOGENETIC PROTEINS (BMPs)**	BMPs are involved in all stages of fracture healing by inducing the differentiation of MSCs into chondrogenic and osteogenic lineages, stimulating angiogenesis, and increasing alkaline phosphatase activity [35]. BMPs can produce heterotopic ossification in muscle tissue, as surrounding mesenchymal lineage progenitor cells in adjacent muscle differentiate into osteoblasts and cause mineral deposition under BMP stimulation [36]. Elevated levels of BMP-2 can also activate osteoclasts, leading to bone resorption [37]. In spine surgery, side effects such as inflammation, bone cysts, and neurological impairment have been reported [38].
**ALTERNATIVE CARRIERS FOR GROWTH FACTOR DELIVERY** *NATURAL POLYMERS: collagen; fibrin; chitosan; hyaluronic acid; gelatin; alginate.* *SYNTHETIC POLYMERS: aliphatic polymers (polylactic acid (PLA), polyglycolic acid (PGA), polylactic-coglycolide (PLGA), poly(e-caprolactone) (PCL), poly-p-dioxanone); copolymers (consisting of glycolide/trimethylene carbonate).* *SYNTHETIC BONE GRAFTS: can be synthesized through solvent casting and particulate leaching (SCPL), freeze drying, thermally induced phase separation (TIPS), gas foaming, electrospinning, hydrogel formation, and additive manufacturing.*	Natural polymers such as collagen, fibrin, chitosan, hyaluronic acid, gelatin, and alginate have clear advantages due to their inherent biocompatibility and bioactivity, but they lack the mechanical properties necessary for load-bearing applications. In addition, they have fixed degradation rates, are difficult to harvest and sterilize, and exhibit batch-to-batch variability. In some cases, they can transmit pathogens and induce an immunogenic response [39]. The use of a natural component such as collagen for this purpose appeared to be very promising due to its biodegradability, biocompatibility, and ability to support mineralization and cell ingrowth in an osteoconductive manner [40]. However, there are several serious side effects in clinical applications associated with this carrier system. Placement of the absorbable collagen sponges (ACS) during surgery is often difficult, and, in some cases, secondary displacement of the collagen sponge occurs [41].
**SMALL MOLECULES AS REGULATORS OF BONE MASS** Parathyroid hormone (PTH), KUR-111, KUR-112, and KUR-113.	PTH was initially tested in femur and humerus defects of female sheep, where it was shown to be both osteoconductive and osteoinductive [42]. KUR-113 was developed for fractures at risk of incomplete healing. It was initially tested in tibial shaft fractures in a Phase II clinical trial and later for spinal fusion in patients with degenerative disc disease [43].

**Table 3 ijms-23-03352-t003:** Strategies to heal bone defects using BGS and stem cells (cell-based constructs).

Strategy	Advantages and Disadvantages
Bone marrow stromal cells (BMSCs)	Injection of BMSCs into a stabilized fracture appears to contribute to direct ossification [52].
Adipose-derived mesenchymal cells (ASCs)	In preclinical studies, ASCs were seeded onto scaffolds for critical-size mouse calvarial defects, demonstrating significant intramembranous bone formation and areas of complete bone regeneration, with a contribution of 84–99% from the implanted cells [53]. ASCs have also shown promising results for spinal fusion, with lower infiltration of inflammatory cells along with superior fusion compared to scaffolds without ASCs [54,55].
Periosteum-derived stem cells (PDSCs)	Under serum-containing conditions, human PDSCs (hPDSCs) have shown an in vitro expansion potential of up to 30 population doublings, with cells showing a fibroblast-like morphology and a population doubling time of approximately 55 h [56].
Three-dimensional bioprinting of hydrogels and cells or bioactive molecules	The tricalcium phosphate (TCP) scaffold was able to specifically enhance the expression of genes of the osteoclast differentiation pathway and the extracellular space to promote osteoclast differentiation and favor the process of bone remodeling [57].

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
