# Peer review of "Bone Healing Materials in the Treatment of Recalcitrant Nonunions and Bone Defects"

_ijms, 2022, doi:10.3390/ijms23063352_

Round 1

Reviewer 1 Report

The tile of this review should be more specific.

In the abstract, the authors did not summarize the main focus of this review.

The contents of all the tables and figures are too simple. More contents should be provided and summarized with relative references.

For each type of bone healing materials (e.g., BGS, BGS combined with bioactive molecules, Bone graft substitutes and stem cells, and bone adhesives), their diagrammatic sketch of these materials should be provided as figures.

The advantages and disadvantages of each type of materials should be thoroughly discussed.

The therapy mechanisms of each type of materials should be discussed, preferably providing a few figures to elucidate the mechanisms.

Author Response

The title of this review should be more specific.

AUTHOR: The title has been changed as follows: “Bone healing materials in the treatment of recalcitrant nonunions and bone defects”

In the abstract, the authors did not summarize the main focus of this review.

AUTHOR: The “abstract” has been rewritten (IN RED).

The contents of all the tables and figures are too simple.

AUTHOR: I have changed the tables and figure to make them more illustrative (not too simple).

More contents should be provided and summarized with relative references.

AUTHOR: I do not know what contents the Reviewer is meaning.

For each type of bone healing materials (e.g., BGS, BGS combined with bioactive molecules, Bone graft substitutes and stem cells, and bone adhesives), their diagrammatic sketch of these materials should be provided as figures.

AUTHOR: It is very difficult to show these concepts in figures. These concepts have been explained within the text.

The advantages and disadvantages of each type of materials should be thoroughly discussed.

AUTHOR: I have emphasized the advantages and disadvantages of each type of material (IN RED in the text), and in the new TABLES.

The therapy mechanisms of each type of materials should be discussed, preferably providing a few figures to elucidate the mechanisms.

AUTHOR: The therapy mechanism of each type of material has been discussed within the text. It is very difficult to show such a mechanism in a figure.

Reviewer 2 Report

The manuscript “Review of Recent Developments in Bone Healing Materials” present an interesting review but need major corrections before publication

The manuscript present interesting data but tables and figures do not highlight the information presented they need to be improved. A better schematic presentation is needed. The manuscript is too long the informations presented should be better synthetize. A lot of unnecessary detail has been included and should be deleted to better capture the reader's interest.

The conclusion is out of the scope of the review and should be rewritten

Author Response

The manuscript “Review of Recent Developments in Bone Healing Materials” presents an interesting review but need major corrections before publication

The manuscript present interesting data but tables and figures do not highlight the information presented they need to be improved. A better schematic presentation is needed.

AUTHOR: I have changed tables and figure, and made a better schematic presentation.

The manuscript is too long the information presented should be better synthetized. A lot of unnecessary detail has been included and should be deleted to better capture the reader's interest.

AUTHOR: I have shortened the manuscript a lot (around 50%).

The conclusion is out of the scope of the review and should be rewritten

AUTHOR: I have rewritten the “conclusion” (IN BLUE)

Round 2

Reviewer 1 Report

The manuscript can be accepeted now.

Reviewer 2 Report

The author realy improve the quality of the manuscript. The revised version is acceptable in the present form.